# A cortical network processes auditory error signals during human speech production to maintain fluency

**Muge Ozker**[1]*, **Werner Doyle**[2], **Orrin Devinsky**[1], **Adeen Flinker**[1,3]

**1** Department of Neurology, New York University School of Medicine, New York, New York, United States of America, **2** Department of Neurosurgery, New York University School of Medicine, New York, New York, United States of America, **3** Department of Biomedical Engineering, New York University School of Engineering, New York, New York, United States of America

* mozker@gmail.com

**Data Availability Statement:** All behavioral and neural data files are available from Flinker Lab Github repository (https://github.com/flinkerlab/DelayedAuditoryFeedback).

## Abstract

Hearing one's own voice is critical for fluent speech production as it allows for the detection and correction of vocalization errors in real time. This behavior known as the auditory feedback control of speech is impaired in various neurological disorders ranging from stuttering to aphasia; however, the underlying neural mechanisms are still poorly understood. Computational models of speech motor control suggest that, during speech production, the brain uses an efference copy of the motor command to generate an internal estimate of the speech output. When actual feedback differs from this internal estimate, an error signal is generated to correct the internal estimate and update necessary motor commands to produce intended speech. We were able to localize the auditory error signal using electrocorticographic recordings from neurosurgical participants during a delayed auditory feedback (DAF) paradigm. In this task, participants hear their voice with a time delay as they produced words and sentences (similar to an echo on a conference call), which is well known to disrupt fluency by causing slow and stutter-like speech in humans. We observed a significant response enhancement in auditory cortex that scaled with the duration of feedback delay, indicating an auditory speech error signal. Immediately following auditory cortex, dorsal precentral gyrus (dPreCG), a region that has not been implicated in auditory feedback processing before, exhibited a markedly similar response enhancement, suggesting a tight coupling between the 2 regions. Critically, response enhancement in dPreCG occurred only during articulation of long utterances due to a continuous mismatch between produced speech and reafferent feedback. These results suggest that dPreCG plays an essential role in processing auditory error signals during speech production to maintain fluency.

## Introduction

Human speech production is strongly influenced by the auditory feedback it generates. When we speak, we continuously monitor our vocal output and adjust our vocalization to maintain fluency. For example, speakers involuntarily raise their voice to be more audible when auditory

**Funding:** This study was supported by grants from the NIH (F32 DC018200 Ruth L. Kirschstein postdoctoral fellowship from the National Institute on Deafness and Other Communication Disorders to M.O. and R01NS109367 from the National Institute of Neurological Disorders and Stroke to A. F.) and the NSF (CRCNS 1912286 to A.F.) and by the Leon Levy Foundation Fellowship (to M.O.). The funders had no role in study design, data collection and analysis, decision to publish, or preparation of the manuscript.

**Competing interests:** "The authors have declared that no competing interests exist."

**Abbreviations:** CT, computed tomography; DAF, delayed auditory feedback; dPreCG, dorsal precentral gyrus; DTW, dynamic time warping; ECoG, electrocorticography; EEG, electroencephalography; FDR, false discovery rate; IFG, inferior frontal gyrus; MNI, Montreal Neurological Institute; NMF, nonnegative matrix factorization; postCG, postcentral gyrus; SMG, supramarginal gyrus; STG, superior temporal gyrus; TTL, transistor–transistor logic; vPreCG, ventral precentral gyrus.

feedback is masked in the presence of background noise [1,2]. Similarly, when speakers hear themselves with a delay (e.g., voice delays or echoes in teleconferencing), they compensate for the auditory feedback delay by slowing down and resetting their speech. This compensatory adjustment of human vocalization provides evidence for a mechanism that detects and corrects vocal errors in real time. Abnormal auditory feedback control has been implicated in various disorders including stuttering, aphasia, Parkinson disease, autism spectrum disorder, and schizophrenia [3–7]; however, the neural underpinnings of this dynamic system remain poorly understood.

Predictive models of speech motor control suggest that the brain generates an internal estimate of the speech output during speech production using an efference copy of the vocal motor command. When there is a mismatch between this internal estimate and the perceived (reafferent) auditory feedback, the auditory response is enhanced to encode the mismatch. This auditory error signal is then relayed to vocal motor regions for the real-time correction of vocalization in order to produce the intended speech [8–10].

In support of these models, electrophysiological studies in nonhuman primates demonstrated increased activity in auditory neurons when the frequency of the auditory feedback is shifted during vocalization [11]. Behavioral evidence in human studies showed that when formant frequencies of a vowel or the fundamental frequency (pitch) is shifted, speakers change their vocal output in the opposite direction of the shift to compensate for the spectral perturbation [12–14]. In line with nonhuman primate studies, human neurosurgical recordings as well as neuroimaging studies demonstrated that these feedback-induced vocal adjustments are accompanied by enhanced neural responses in auditory regions [15–17].

An alternative method to manipulating the spectral features of auditory feedback is altering its temporal features by delaying the feedback of the voice in real time, termed "delayed auditory feedback (DAF)." First described in the 1950s, DAF strongly disrupts speech fluency leading to slower speech rate, pauses, syllable repetitions, and increased voice pitch or intensity [18–20]. Further, higher susceptibility to DAF occurs in autism spectrum disorder, nonfluent primary progressive aphasia, schizophrenia, and other neurological disorders [4–6]. Interestingly, DAF improves speech fluency in individuals who stutter and is a therapeutic approach in speech therapy for stuttering and Parkinson disease [21–23]. Although these behavioral effects have been widely studied in both normal and clinical groups, only a few neuroimaging studies have investigated the neural responses. Studies have demonstrated enhanced responses in bilateral posterior superior cortices during delayed feedback compared with normal auditory feedback conditions [24–26]. However, the exact temporal dynamics and spatial distribution of the cortical networks underlying speech production and reafferent feedback processing remain unknown.

To address this issue, we leveraged the excellent spatial and temporal resolution of electrocorticography (ECoG). Using ECoG, we acquired direct cortical recordings from 15 epilepsy patients while they read aloud words and sentences. As they spoke, we recorded their voice and played it back to them through earphones either simultaneously (no delay) or with a delay (50, 100, and 200 milliseconds). Behaviorally, we found that participants slowed down their speech to compensate for the delay and did so more profoundly when producing sentences. Neurally, there was a significant increase in activity across a large speech network encompassing temporal, parietal, and frontal sites that scaled with the duration of feedback delay. Critically, when speech was slowed down and became effortful, the dorsal division of the precentral gyrus was preferentially recruited at an early timing to support ongoing articulation. To our knowledge, we introduce the first temporally delayed feedback processing investigation with invasive human electrophysiology in which we reveal the fine-grained spatiotemporal dynamics of the neural mechanisms underlying compensatory adjustment of human vocalization.

## Materials and methods

### Participant information

All experimental procedures were approved by the New York University School of Medicine Institutional Review Board, which operates in accordance with the principles expressed in the Declaration of Helsinki. A total of 15 neurosurgical epilepsy patients (8 females, mean age: 34, 2 right, 9 left, and 4 bilateral hemisphere coverage) implanted with subdural and depth electrodes provided informed consent to participate in the research protocol. All consent was obtained in writing and then requested again orally prior to the beginning of the experiment. Electrode implantation and location were guided solely by clinical requirements. A total of 3 participants were consented separately for higher density clinical grid implantation, which provided denser sampling of underlying cortex.

### Experiment setup

Participants were tested while resting in their hospital beds in the epilepsy monitoring unit. Visual stimuli were presented on a laptop screen positioned at a comfortable distance from the participant. Auditory stimuli were presented through earphones (Bed Phones On-Ear Sleep Headphones Generation 3, Dubs Concepts LLC, Oregon, USA), and participants' voice was recorded using an external microphone (Zoom H1 Handy Recorder, Zoom Corporation, Tokyo, Japan).

### DAF experiment

The experiment consisted of a word reading session and a sentence reading session. A total of 10 different 3-syllable words (e.g., document) were used in the word reading session, and 6 different 8-word sentences (e.g., the cereal was fortified with vitamins and nutrients) were used in the sentence reading session. Text stimuli were visually presented on the screen, and participants were instructed to read them out loud. As participants spoke, their voices were recorded using the laptop's internal microphone, delayed at 4 different amounts (no delay, 50, 100, and 200 milliseconds) using custom script (MATLAB, Psychtoolbox-3, The MathWorks Inc., Massachusetts, USA) and played back to them through earphones. A transistor–transistor logic (TTL) pulse marking the onset of a stimulus, the delayed feedback voice signal (what the participant heard), and the actual microphone signal (what the participant spoke) were fed in to the electroencephalography (EEG) amplifier as an auxiliary input in order to acquire them in sync with the EEG samples. Trials with different amount of feedback delays (18 to 60 repetitions for each delay) were presented randomly with at least a 1-second intertrial interval.

### ECoG recording

ECoG was recorded from implanted subdural platinum–iridium electrodes embedded in flexible silicon sheets (2.3-mm diameter exposed surface, 8 × 8 grid arrays, and 4 to 12 contact linear strips, 10-mm center-to-center spacing, Ad-Tech Medical Instrument, Racine, Wisconsin, USA) and penetrating depth electrodes (1.1-mm diameter, 5- to 10-mm center-to-center spacing 1 × 8 or 1 × 12 contacts, Ad-Tech Medical Instrument). Three participants consented to a research hybrid grid implanted that included 64 additional electrodes between the standard clinical contacts (16 × 8 grid with sixty-four 2-mm macro contacts at 8 × 8 orientation and sixty-four 1-mm micro contacts in between, providing 10-mm center-to-center spacing between macro contacts and 5-mm center-to-center spacing between micro/macro contacts, PMT, Chanhassen, Minnesota, USA). Recordings were made using 1 of 2 amplifier types: NicoletOne amplifier (Natus Neurologics, Middleton, Wisconsin, USA), band-pass filtered from 0.16 to 250 Hz and digitized at 512 Hz. Neuroworks Quantum Amplifier (Natus

Biomedical, Appleton, Wisconsin, USA) recorded at 2,048 Hz, band-pass filtered at 0.01 to 682.67 Hz, and then downsampled to 512 Hz. A 2-contact subdural strip facing toward the skull near the craniotomy site was used as a reference for recording, and a similar 2-contact strip screwed to the skull was used for the instrument ground. ECoG and experimental signals (trigger pulses that mark the appearance of visual stimuli on the screen, microphone signal from speech recordings, and auditory playback signal that was heard by the participants through earphones) were acquired simultaneously by the EEG amplifier in order to provide a fully synchronized data set. Recorded microphone and feedback signals were analyzed to ensure that the temporal delay manipulation by our MATLAB code produced the intended delay.

## Electrode localization

Electrode localization in participant space as well as the Montreal Neurological Institute (MNI) space was based on coregistering a preoperative (no electrodes) and postoperative (with electrodes) structural MRI (in some cases a postoperative computed tomography (CT) was employed depending on clinical requirements) using a rigid body transformation. Electrodes were then projected to the surface of cortex (preoperative segmented surface) to correct for edema induced shifts following previous procedures [27] (registration to the MNI space was based on a nonlinear DARTEL algorithm [28]). Within-participant anatomical locations of electrodes were based on the automated FreeSurfer segmentation of the participant's preoperative MRI. All middle and caudal superior temporal gyrus (STG) electrodes were grouped as STG, and all pars opercularis and pars triangularis electrodes were grouped as inferior frontal gyrus (IFG) electrodes. Precentral electrodes with a z coordinate smaller than ±40 were grouped as ventral precentral gyrus (vPreCG), and those with an z coordinate larger than or equal to ±40 were grouped as dorsal precentral gyrus (dPreCG) together with electrodes located in caudal middle frontal gyrus.

## Neural data analysis

A common average reference was calculated by subtracting the average signal across all electrodes from each individual electrode's signal (after rejection of electrodes with artifacts caused by line noise, poor contact with cortex, and high amplitude shifts). Continuous data streams from each channel were epoched into trials (from −1.5 seconds to 3.5 seconds for word stimuli and from −1.5 seconds to 5.5 seconds for sentence stimuli with respect to speech onset). Line noise at 60, 120, and 180 Hz were filtered out, and data were transformed to time frequency space using the multitaper method (MATLAB, FieldTrip toolbox) with 3 Slepian tapers; frequency windows from 10 to 200 Hz; frequency steps of 5 Hz; time steps of 10 milliseconds; temporal smoothing of 200 milliseconds; and frequency smoothing of ±10 Hz. The high gamma broadband response (70 to 150 Hz) at each time point following stimulus onset was measured as the percent signal change from baseline, with the baseline calculated over all trials in a time window from −500 to −100 milliseconds before stimulus onset. High gamma response duration for each electrode was measured by calculating the time difference at full width quarter maximum of the response curve.

## Electrode selection

We recorded from a total of 1,693 subdural and 608 depth electrode contacts in 15 participants. Electrodes were examined for speech-related activity defined as significant high gamma broadband responses. For DAF word reading tasks, electrodes that showed significant response increase ($p < 10^{-4}$, unpaired $t$ test) either before (−0.5 to 0 seconds) or after speech

onset (0 to 0.5 seconds) with respect to a baseline period (−1 to −0.6 seconds) and at the same time had a large signal-to-noise ratio ($\mu/\sigma > 0.7$) during either of these time windows were selected. For the DAF sentence reading task, the same criteria were applied, except the time window after speech onset was longer (0 to 3 seconds). Electrode selection was first performed for each task separately, and then electrodes that were commonly selected for both tasks were further analyzed.

## Clustering analysis

Nonnegative matrix factorization (NMF) was used to identify major response patterns across different brain regions during speech production. NMF is an unsupervised dimensionality reduction (or clustering) technique that reveals the major patterns in the data without specifying any features [29]. We performed the clustering analysis using the data from the DAF word reading task, since this data set contained a large number of trials. We combined responses from all participants by concatenating trials and electrodes forming a large data matrix A (electrodes by time points). Matrix A was factorized into 2 matrices W and H by minimizing the root mean square residual between A and W*H (nnmf function in MATLAB). Factorization was performed based on 2 clusters to represent the 2 major predicted speech-related components in the brain: auditory and motor.

## DTW analysis

In the sentence reading task, 6 different sentences (e.g., sentence #1: "The cereal was fortified with vitamins and nutrients") were presented. Dynamic time warping (DTW) analysis was performed separately for the 6 different sentence stimuli. First, the speech spectrogram was averaged across frequencies for each sentence stimuli. Then, the mean spectrograms were averaged across trials of the same sentence stimuli (e.g., trials in which sentence #1 was presented with no delay). Then, DTW is performed to compare the averaged spectrograms for no delay and 200-millisecond delay conditions (e.g., sentence #1 with no delay versus sentence #1 with 200-millisecond delay), and the resulting warping paths were applied to the neural response signal for each trial. Finally, the transformed neural responses were averaged across trials for each sentence stimuli. This procedure was performed to compare 2 conditions that resulted in the largest neural response difference (no delay versus 200-millisecond delay).

## Statistical analysis

The effect of DAF on speech behavior was determined by performing a 1-way ANOVA using articulation duration of words and sentences as the dependent variable and delay condition as the independent variable. Participants were introduced as a factor to account for repeated measures. To determine a significant difference in the amplitude of neural response between conditions, the average high gamma activity in a specified time window was compared by performing 1-way ANOVA across all trials in all electrodes using delay condition as the independent variable. Similarly, a significant difference in the duration of neural response was determined by performing 1-way ANOVA across participants using response duration as the dependent variable and delay condition as the independent variable. To assess the sensitivity of a recording location to DAF, Spearman correlation between the neural response and delay condition was calculated for each electrode. To compare DAF sensitivity for word and sentence reading tasks, sensitivity indices of electrodes were compared using a paired *t* test. To reveal how response enhancement to DAF changed across time during the sentence reading task, either 1-way ANOVA or paired *t* test was performed at each time point. Multiple comparisons were then corrected using the false discovery rate (FDR) method [30].

## Results

Participants (*N* = 15) performed a word reading task (single 3-syllable words) while the auditory feedback of their voice was played back to them through earphones either simultaneously (no delay) or with delay (50, 100, and 200 milliseconds), a paradigm known as DAF. We first analyzed the voice recordings of participants and measured the articulation duration at different amount of delays to establish the behavioral effect of DAF (**Fig 1A**). Articulation duration increased slightly with delay: Average articulation duration across participants was 0.698, 0.726, 0.737, and 0.749 milliseconds for no delay, 50-, 100-, and 200-millisecond delay conditions, respectively (**Fig 1B, ANOVA:** F = 7.76, *p* = 0.015).

To quantify the neural response, we used the high gamma broadband signal (70 to 150 Hz; see Materials and methods), a widely used index of cortical activity which correlates with underlying neuronal spike rates [31–33]. Two response patterns emerged among the electrodes that showed significant activity during speech production (see Electrode selection in Materials and methods). In the first pattern, shown on a representative auditory electrode located in the STG (**Fig 1C**), neural response started after speech onset and its amplitude

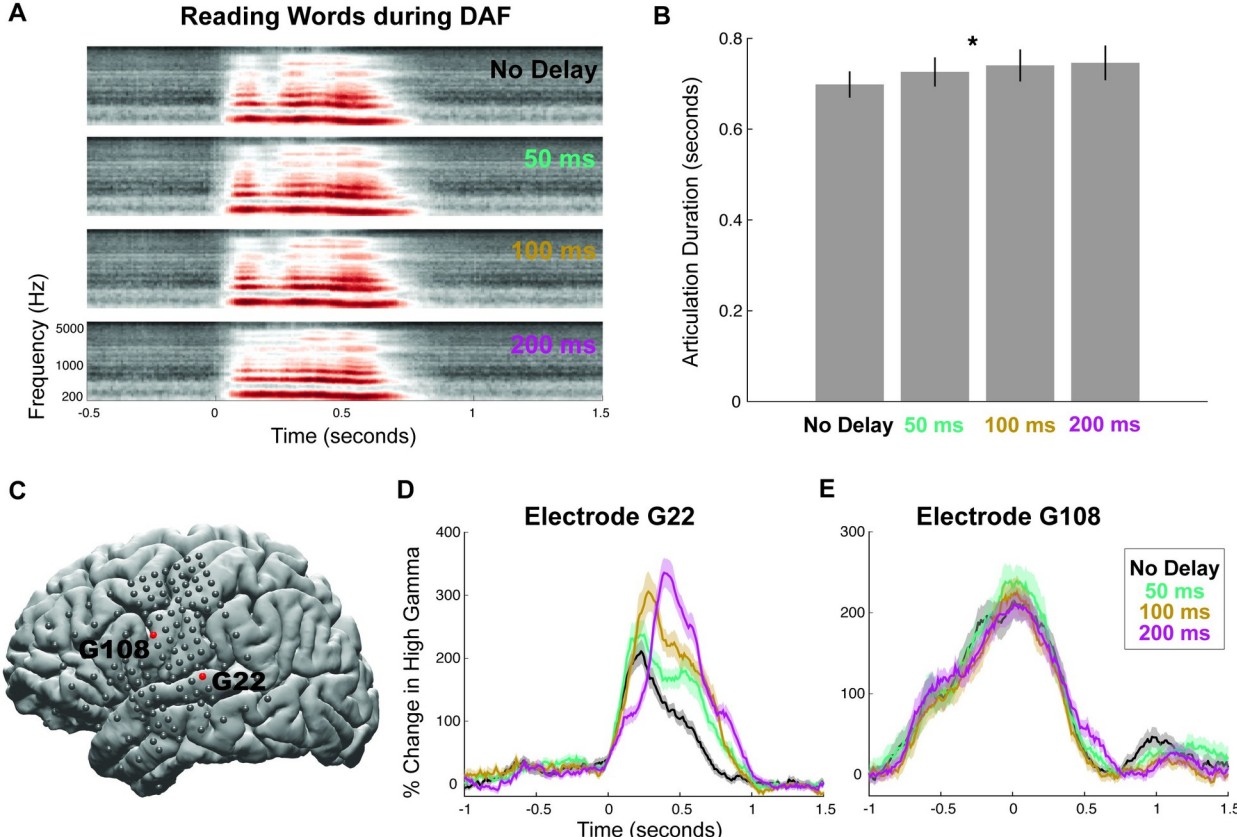

**Fig 1. Behavioral and neural responses during word reading with DAF.** (A) Speech spectrogram of a single participant articulating words during DAF conditions. (B) Mean articulation duration of words during DAF conditions averaged across participants. Error bars show SEM over participants. (C) Cortical surface model of the left hemisphere brain of a single participant. Gray circles indicate the implanted electrodes. Red highlighted electrodes are located on the STG (G22) and on the vPreCG (G108). (D) High gamma responses in an auditory electrode (G22) to articulation of words during DAF conditions (color coded). Shaded regions indicate SEM over trials. (E) High gamma responses in a motor electrode (G108) to articulation of words during DAF conditions (color coded). Shaded regions indicate SEM over trials. The underlying data can be found in https://github.com/flinkerlab/DelayedAuditoryFeedback. DAF, delayed auditory feedback; SEM, standard error of the mean; STG, superior temporal gyrus; vPreCG, ventral precentral gyrus.

increased significantly with delay (**Fig 1D, ANOVA:** F = 37, $p = 1.55 \times 10^{-8}$). In the second pattern, shown on a representative motor electrode located in vPreCG (**Fig 1C**), neural response started before speech onset, and its amplitude was not affected by delay (**Fig 1E, ANOVA:** F = 0.084, $p = 0.772$). This result demonstrated that DAF affected the neural response in auditory sites that are involved in speech processing.

To characterize the 2 major response patterns in the brain, we chose to use an unbiased, data-driven approach that does not impose any assumptions or restrictions on the selection of responses. We performed an unsupervised clustering analysis using the NMF algorithm on neural responses across all delay conditions, brain sites, and participants [29,34]. The clustering analysis identified the major response patterns represented by 2 distinct clusters, which corroborated our representative results shown in a single participant (**Fig 1C–1E**) as well as visual inspection of the data across participants. The first response pattern (Cluster 1, $N = 125$ electrodes) started after speech onset and peaked at 320 milliseconds reaching 115% change in amplitude. The second response pattern (Cluster 2, $N = 253$ electrodes) started much earlier approximately 750 milliseconds prior to speech onset and peaked 140 milliseconds after speech onset reaching 60% change in amplitude (**Fig 2A**). These 2 clusters had a distinct anatomical distribution (**Fig 2B**): Cluster 1 was mainly localized to STG, suggesting an auditory

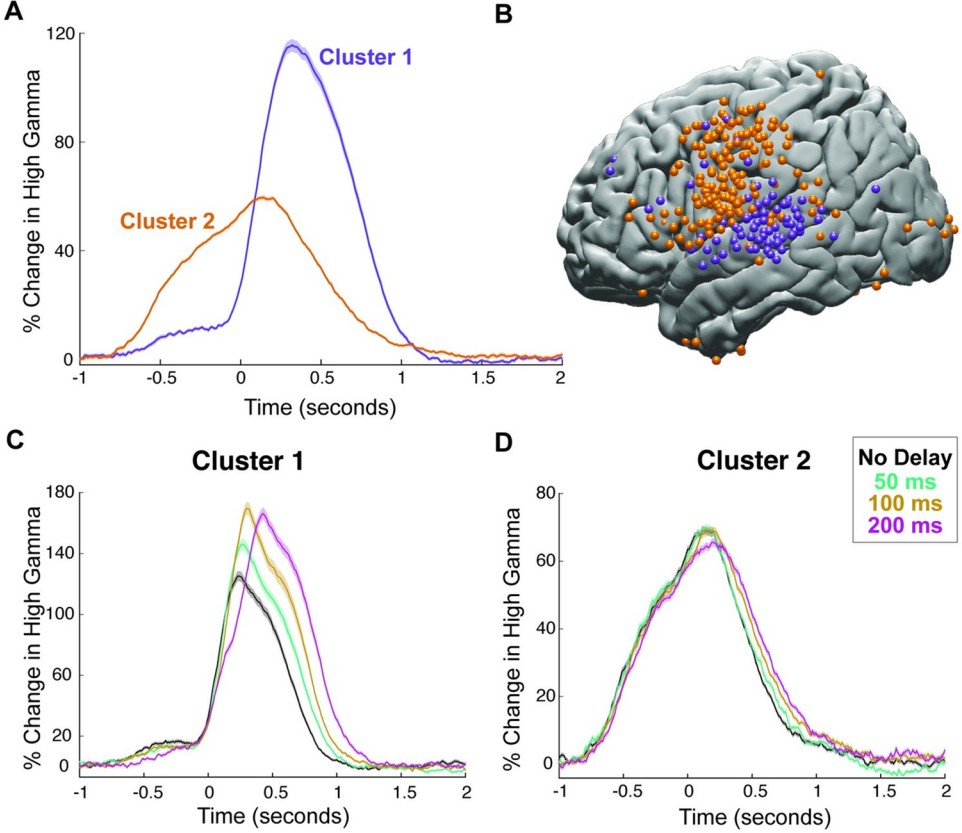

**Fig 2. Clustering with NMF. (A)** High gamma responses averaged across electrodes in the 2 clusters provided by the unsupervised NMF. Shaded regions indicate SEM over trials. **(B)** Spatial distribution on cortex of electrodes in the 2 clusters displayed on the left hemisphere of a template brain. **(C)** High gamma responses to articulation of words during DAF conditions averaged across electrodes in Cluster 1. Shaded regions indicate SEM over trials. **(D)** High gamma response to articulation of words during DAF averaged across electrodes in Cluster 2. Shaded regions indicate SEM over trials. The underlying data can be found in https://github.com/flinkerlab/DelayedAuditoryFeedback. DAF, delayed auditory feedback; NMF, nonnegative matrix factorization; SEM, standard error of the mean.

function, while Cluster 2 was localized to frontal cortices, suggesting a premotor and motor function.

Next, we examined the effect of DAF on these 2 clusters. The amplitude of the neural response increased significantly with delay in Cluster 1 (**Fig 2C, ANOVA:** F = 5.35, $p$ = 0.02), but not in Cluster 2 (**Fig 2D, ANOVA:** F = 1.65, $p$ = 0.2). The duration of the neural response did not show a significant increase in either of the clusters (**ANOVA:** F = 1, $p$ = 0.32 for Cluster 1 and F = 0.01, $p$ = 0.92 for Cluster 2).

Reading words with DAF slightly prolonged articulation duration, and while it increased neural responses in auditory regions, it did not affect responses in motor regions. We hypothesized that a longer and more complex stimulus may elicit a stronger behavioral response and motor regions will show an effect of DAF when articulation is strongly affected. To test this prediction, we performed another experiment in which participants read aloud sentences during DAF. Indeed, articulating longer speech segments (8-word sentences) during DAF resulted in a significantly stronger behavioral effect (**Fig 3A**; see S1 Text for a detailed speech error analysis). Articulation duration increased significantly with delay: Average articulation duration across participants was 2.761, 2.942, 3.214, and 3.418 seconds for no delay, 50-, 100-, and 200-millisecond delay conditions, respectively (**Fig 3B, ANOVA:** F = 20.54, $p$ = 0.0005).

Next, we examined the neural response to DAF in a representative auditory electrode located in the STG (**Figs 1C and 3C**) and found that neural response amplitude increased significantly with delay (**Fig 3D, ANOVA:** F = 48.28, $p$ = $7.8 \times 10^{-17}$). In a representative motor electrode located in vPreCG (**Figs 1C and 3C**), neural response started before speech onset, and its amplitude was not affected by delay (**Fig 3E, ANOVA:** F = 0.29, $p$ = 0.83). We then examined the neural response in the 2 electrode clusters we identified previously (**Fig 2B**). When reading words during DAF, amplitude of the neural response increased with delay in Cluster 1 but not in Cluster 2 (**Fig 2C and 2D**). However, when reading sentences during DAF, neural response in both clusters showed a sustained effect (**Fig 3F and 3G, ANOVA:** F = 18, $p$ = $2.95 \times 10^{-5}$ for Cluster 1 and F = 4.8, $p$ = 0.03 for Cluster 2). Also, when reading words during DAF, duration of the neural response in neither of the clusters showed a significant effect of delay. However, when reading sentences during DAF, neural response duration in both clusters increased significantly with delay paralleling the significant behavioral effect of DAF on articulation duration (**ANOVA:** F = 21.6, $p$ = $10^{-5}$ for Cluster 1 and F = 35.5, $p$ = $10^{-8}$ for Cluster 2).

Our clustering analysis identified 2 response components that were mostly anatomically distinct reflecting an auditory response to self-generated speech and a motor response to articulation. The auditory component was unique in exhibiting an enhanced response during both word reading and sentence reading, with DAF likely representing an auditory error signal. Response enhancement changed as a function of feedback delay, which suggests that auditory error signal does not simply encode the mismatch between intended and perceived speech but is sensitive to the amount of mismatch. To quantify the error signal, we calculated a sensitivity index for each electrode by measuring the trial-by-trial Spearman correlation between the delay condition and the neural response averaged over a 0- to 1-second time window for words and over 0 to 3 seconds for sentences. A large sensitivity value indicated a strong response enhancement with increasing delays.

Comparing sensitivity indices for word reading and sentence reading tasks revealed that several sites such as dPreCD and IFG showed higher DAF sensitivity in the sentence reading task (**Fig 4A and 4B,** S3 **and** S4 **Figs,** S3 **and** S4 **Text**). Distribution of sensitivity indices across all electrodes demonstrated that a larger number of electrodes displayed higher DAF sensitivity for the sentence reading task compared to the word reading task (paired $t$ test: t = 11.15, $p$ = $8.3 \times 10^{-24}$, **Fig 4C**). This result suggests that articulating longer and more complex speech

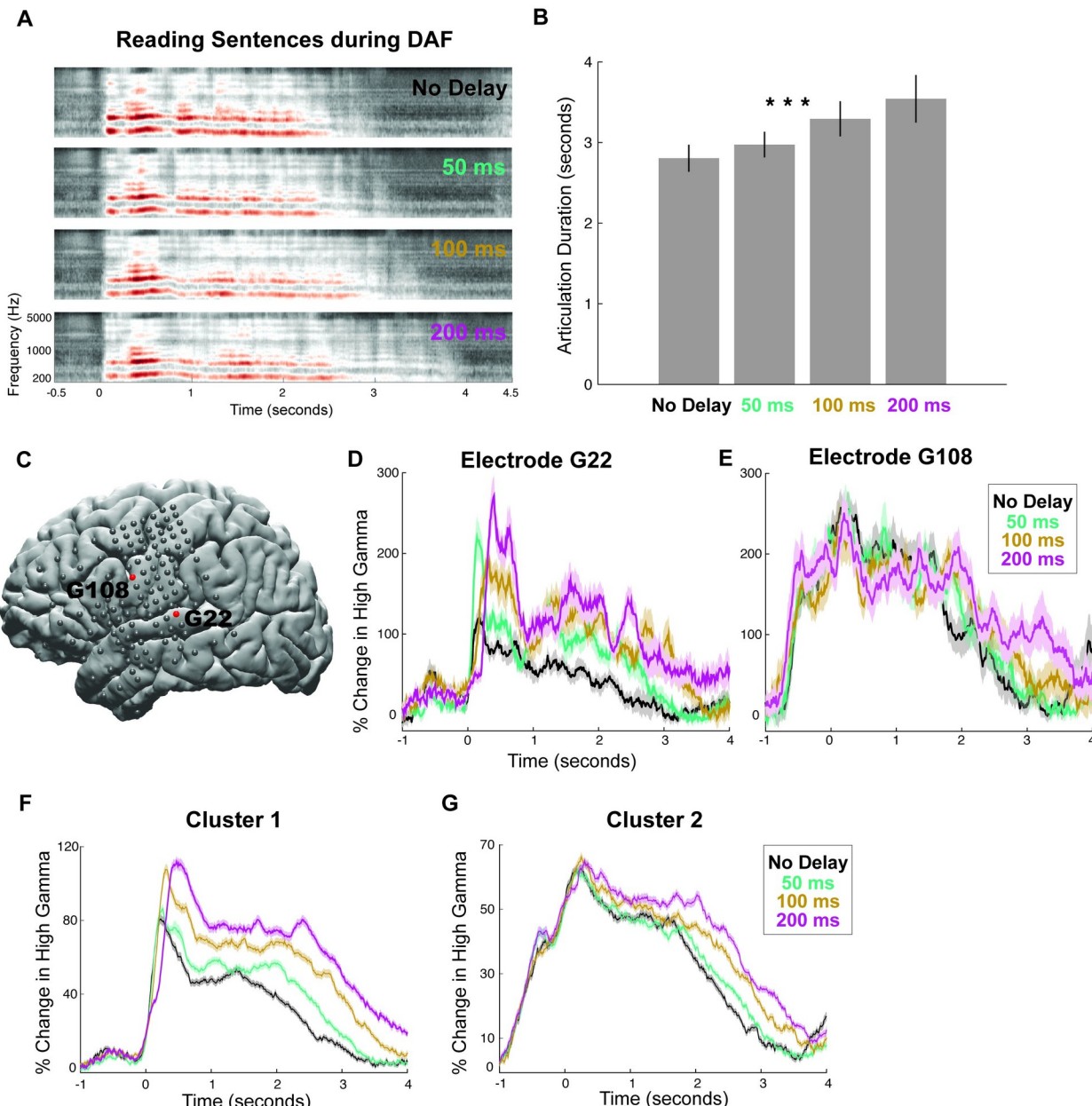

**Fig 3. Behavioral and neural responses during sentence reading with DAF.** **(A)** Speech spectrogram of a single participant articulating sentences during DAF conditions showing a marked increase in articulation duration. **(B)** Mean articulation duration of sentences during DAF conditions averaged across participants showing a significant effect of duration. Error bars show SEM over participants. **(C)** Cortical surface model of the left hemisphere brain of a single participant. Gray circles indicate the implanted electrodes. Red highlighted electrodes are located on the STG (G22) and on the vPreCG (G108). **(D)** High gamma responses in an auditory electrode (G22) to articulation of sentences during DAF conditions (color coded). Shaded regions indicate SEM over trials. **(E)** High gamma responses in a motor electrode (G108) to articulation of sentences during DAF conditions (color coded). Shaded regions indicate SEM over trials. **(F)** High gamma responses to articulation of sentences during DAF conditions averaged across electrodes in Cluster 1. Shaded regions indicate SEM over trials. **(G)** High gamma responses to articulation of sentences during DAF conditions averaged across electrodes in Cluster 2. Shaded regions indicate SEM over trials. The underlying data can be found in https://github.com/flinkerlab/DelayedAuditoryFeedback. DAF, delayed auditory feedback; SEM, standard error of the mean; STG, superior temporal gyrus; vPreCG, ventral precentral gyrus.

stimuli during DAF not only elicits a stronger behavioral response but also results in stronger neural response enhancement across auditory and motor regions and engages a larger brain network uniquely recruiting additional frontal regions.

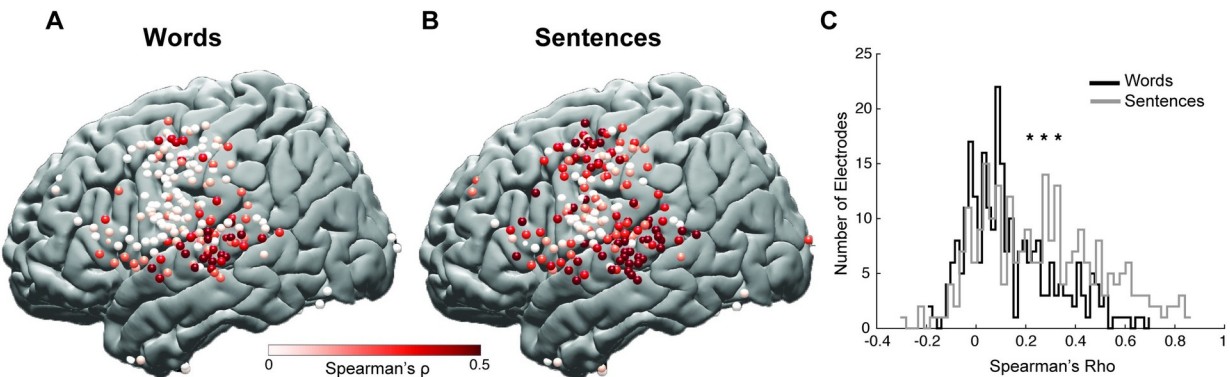

**Fig 4. Neural sensitivity to DAF. (A)** Anatomical map of electrodes across all participants displayed on the left hemisphere of a template brain showing the neural sensitivity to DAF during word reading. **(B)** Anatomical map of electrodes across all participants displayed on the left hemisphere of a template brain showing the neural sensitivity to DAF during sentences reading. **(C)** Histogram showing the distribution of DAF sensitivity indices across electrodes for the word reading and sentence reading tasks. The underlying data can be found in https://github.com/flinkerlab/DelayedAuditoryFeedback. DAF, delayed auditory feedback.

We further examined the neural response to DAF in 6 different regions of interest based on within-participant anatomy: STG, vPreCG, dPreCG, postCG, SMG, and IFG (**Fig 5A–5F, S1 Fig**). Comparing sensitivity indices for word reading and sentence reading tasks in these regions revealed that all 6 regions showed larger sensitivity to DAF during sentence reading (paired $t$ test; STG: t = 6.4, $p = 1.4 \times 10^{-7}$; vPreCG: t = 5.3, $p = 1 \times 10^{-5}$; dPreCG: t = 8.3, $p = 2.7 \times 10^{-9}$; postCG: t = 5, $p = 6.4 \times 10^{-6}$; SMG: t = 2.3, $p = 0.03$; IFG: t = 4.5, $p = 4 \times 10^{-4}$; **Fig 5G**).

To reveal how response enhancement to DAF changed across time during the sentence reading task, we performed a 1-way ANOVA at each time point, corrected multiple comparisons using the FDR method (q = 0.05) and marked the time points when the neural response to the 4 delay conditions were significantly different ($p < 0.001$) for at least 200 consecutive milliseconds. Significant divergence onset during sentence reading started the earliest in STG at 80 milliseconds after speech onset, followed by dPreCG at 360 milliseconds and SMG gyrus at 680 milliseconds, and lasted throughout the stimulus. In postCG, vPreCG, and IFG, responses diverged much later at 1.80, 1.88, and 2.30 seconds, respectively (**Fig 5H, S5 Fig, S5 Text**). In postCG, there was a brief period between 110 and 440 milliseconds at which neural responses were diverged significantly; however, this period did not reflect a neural response enhancement with increasing delays. Altogether, these divergence onsets reveal when cortical regions engage in auditory error processing and provide evidence for 2 distinct timeframes of early (STG, dPreCG, and SMG) and late (postCG, vPreCG, and IFG) recruitment.

Examining different regions of the speech network revealed variable degree of neural response enhancement to DAF. The increase in response amplitude was usually accompanied by an increase in response duration, which was a result of longer articulation duration. In order to disentangle the enhanced neural response representing an error signal from longer articulation duration due to the exerted behavior, we applied a temporal normalization technique. We transformed the neural response time series using DTW so that they would match in time span. DTW measures the similarity between 2 temporal sequences with different lengths by estimating a distance metric (a warping path) that would transform and align them in time. Matching the neural responses in time allowed us to directly compare their amplitudes and identify which brain regions produce an error signal in response to DAF rather than just sustained activity in time due to longer articulation.

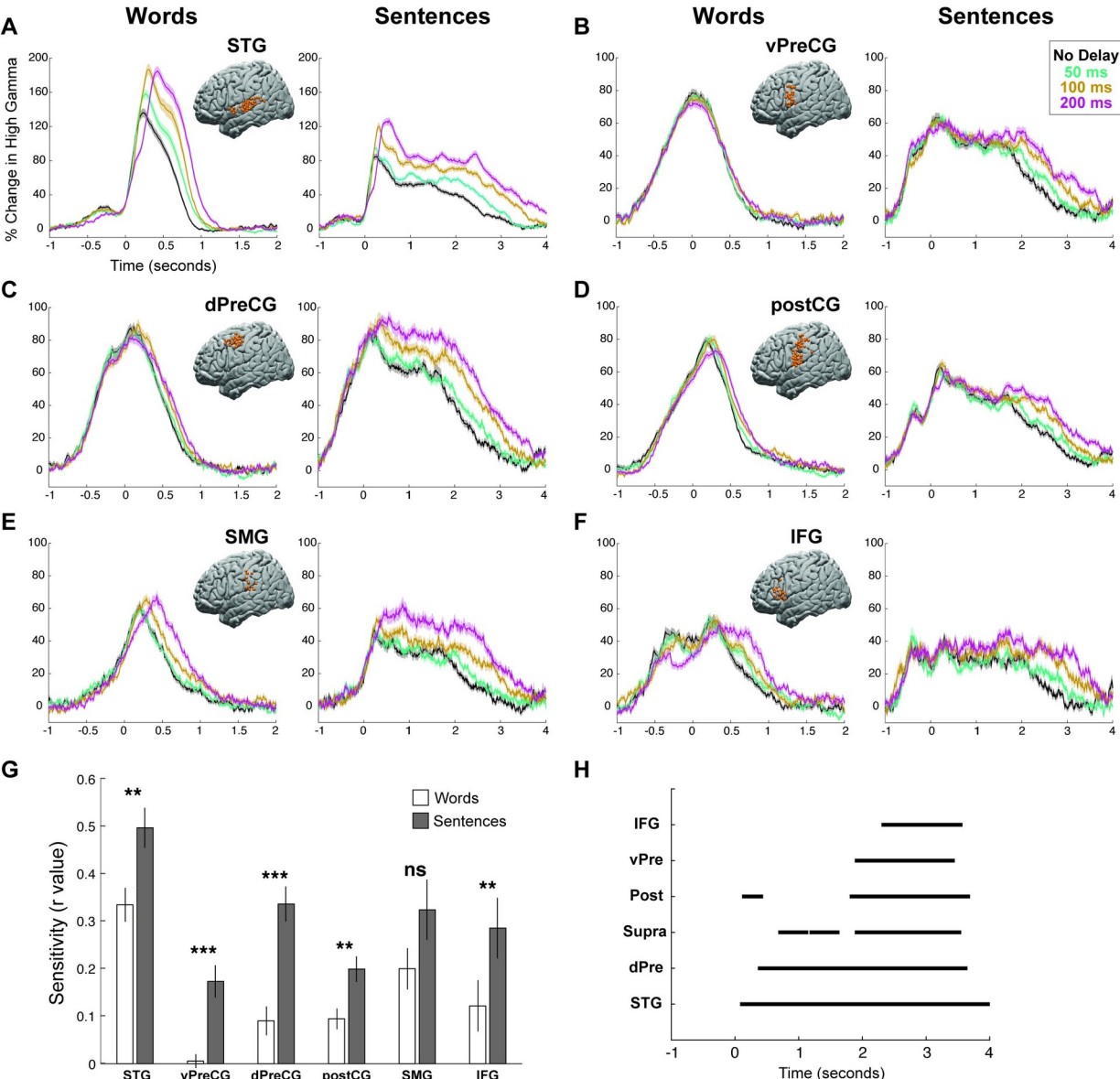

**Fig 5. Neural responses to DAF by regions. (A–F)** High gamma responses to articulation of words and sentences during DAF in 6 different regions: STG (A), vPreCG (B), dPreCG (C), postCG (D), SMG (E), and IFG (F). Inset brain figures shows the location of electrodes across all participants on the left hemisphere of a template brain. Colors represent the various DAF conditions and shaded regions indicate SEM over trials. **(G)** Sensitivity to DAF during word reading and sentence reading tasks averaged across electrodes (error bars indicate SEM over electrodes) in 6 different regions. **(H)** Time intervals when the neural response to reading sentences during DAF diverged significantly across conditions within each of the 6 different regions. The underlying data can be found in https://github.com/flinkerlab/DelayedAuditoryFeedback. DAF, delayed auditory feedback; dPreCG, dorsal precentral gyrus; IFG, inferior frontal gyrus; postCG, postcentral gyrus; SEM, standard error of the mean; SMG, supramarginal gyrus; STG, superior temporal gyrus; vPreCG, ventral precentral gyrus.

We compared 2 conditions, which show the largest neural response difference in terms of amplitude and duration: no delay and 200-millisecond delay conditions (see DTW analysis in Materials and methods). After performing DTW, we compared neural response durations to confirm that the signals are properly aligned in time. Only for STG, the neural response duration was marginally larger for 200-millisecond delay condition after DTW. For the rest of the regions, there was no significant difference in neural response durations confirming a

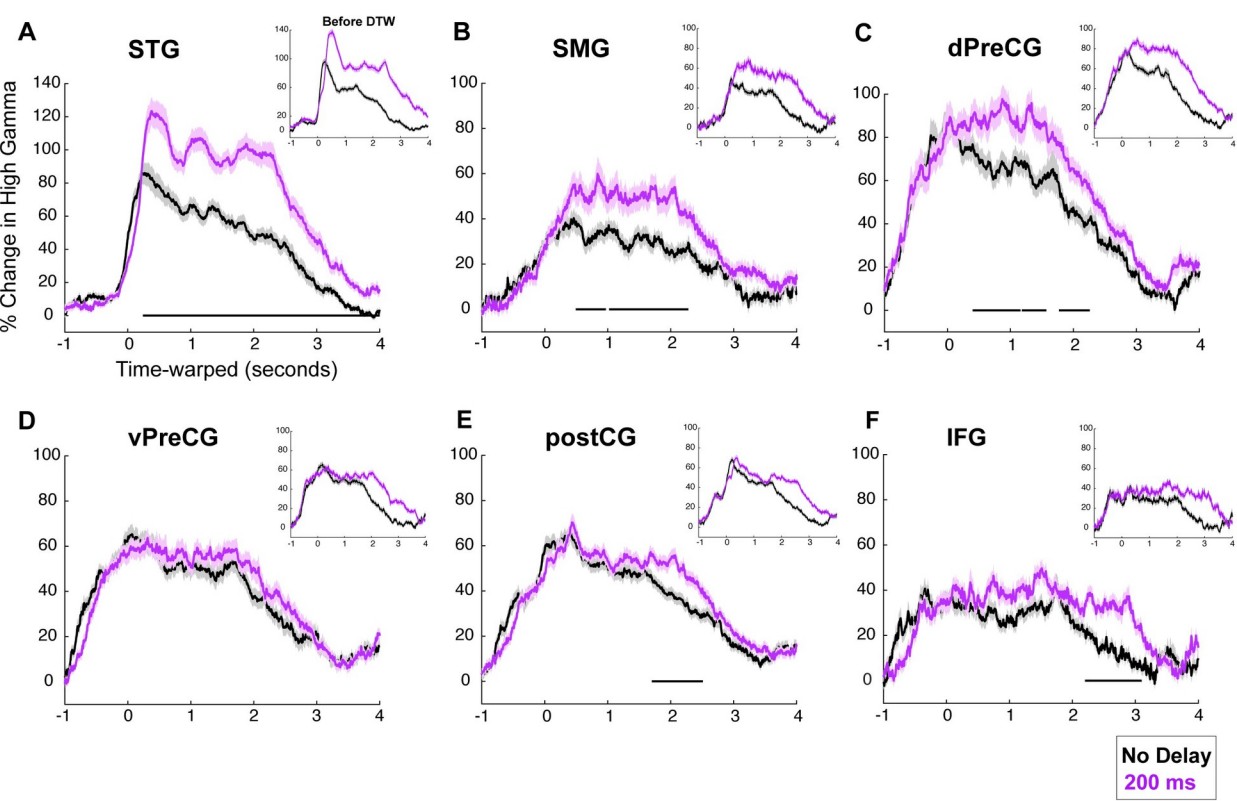

**Fig 6. Time-warped neural responses during sentence reading with DAF. (A–F)** High gamma responses after correction for articulation duration using DTW. Activity locked to articulation of sentences is shown for no delay (black) and 200-millisecond delay (magenta) conditions in 6 different regions: STG, SMG, vPreCG, dPreCG, postCG, and IFG. Inset figures show the uncorrected high gamma responses which include the normal articulation timing. The underlying data can be found in https://github.com/flinkerlab/DelayedAuditoryFeedback. DAF, delayed auditory feedback; dPreCG, dorsal precentral gyrus; DTW, dynamic time warping; IFG, inferior frontal gyrus; postCG, postcentral gyrus; SEM, standard error of the mean; SMG, supramarginal gyrus; STG, superior temporal gyrus; vPreCG, ventral precentral gyrus.

successful alignment (paired *t* test STG: t = 2.2, *p* = 0.03, vPreCG: t = 1.3, *p* = 0.2; dPreCG: t = 1.7, *p* = 0.1; postCG: t = 1.2, *p* = 0.25; SMG: t = 1.6, *p* = 0.14; IFG: t = 0.3, *p* = 0.8).

After aligning the responses in time, to reveal how response enhancement to DAF changed across time, we performed a paired *t* test at each time point with FDR correction (q = 0.05) and marked the time points when the neural response to no delay and 200-millisecond conditions were significantly different (*p* < 0.01) for at least 200 consecutive milliseconds (see Materials and methods). Significant divergence of neural responses started the earliest in STG at 260 milliseconds after speech onset, followed by dPreCG at 400 milliseconds and SMG gyrus at 490 milliseconds. In postCG and IFG, responses diverged much later at 1.70 and 2.20 seconds, respectively. In vPreCG, there was no significant divergence between the neural responses to the 2 delay conditions (**Fig 6A–6F**). Calculating the divergence onsets after aligning the neural responses in time provided further evidence that the error signal centered around 3 major cortical networks: STG, SMG, and dPreCG.

In a DAF paradigm, articulation duration depends on the feedback delay; therefore, dissociating the effect of these 2 factors on the neural response is difficult. In other words, articulation duration becomes longer as the delay is increased, and when articulation duration is longer, the neural response is longer. To dissociate these issues and test whether articulation duration or delay condition leads to enhanced neural responses, we followed another approach in which we either controlled for articulation duration or for the amount of feedback delay and then tested differences in neural responses.

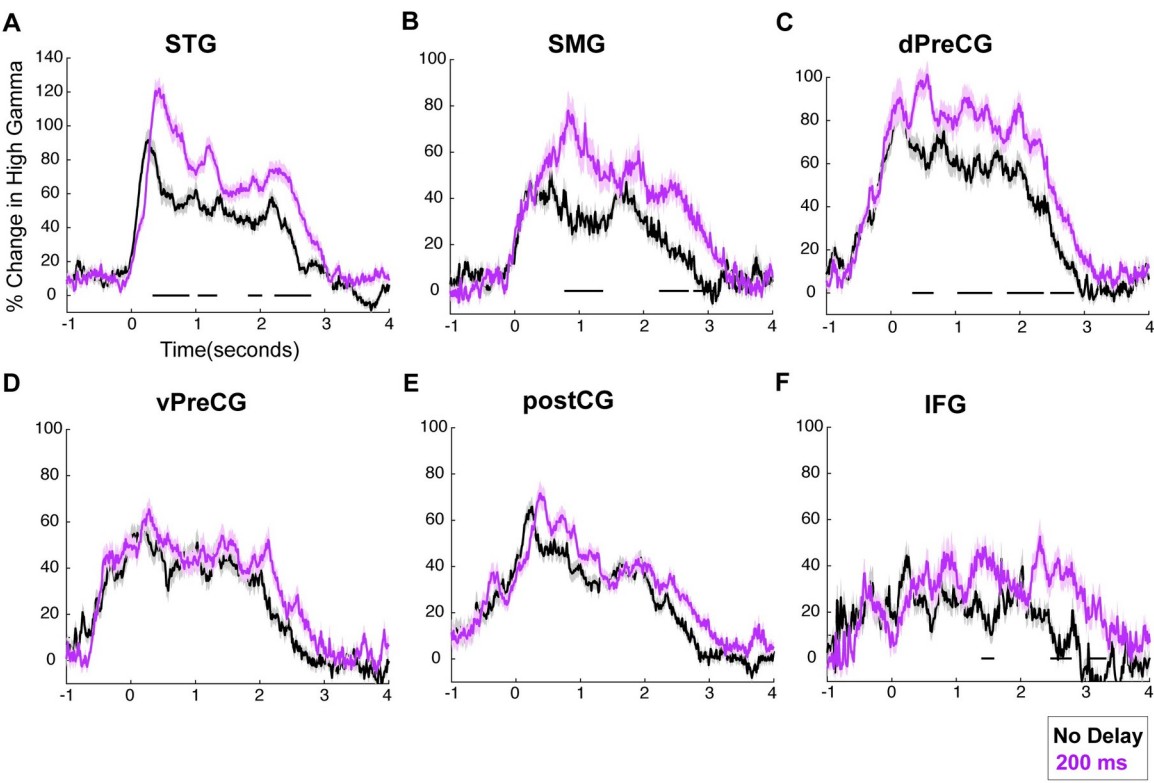

**Fig 7. Neural responses for DAF sentence reading task after controlling for articulation duration. (A–F)** High gamma responses after pairing no delay and 200-millisecond delay condition trials of the same item that match in articulation duration. Activity locked to articulation of sentences is shown for no delay (black) and 200-millisecond delay (magenta) conditions in 6 different regions: STG, SMG, vPreCG, dPreCG, postCG and IFG. The underlying data can be found in https://github.com/flinkerlab/DelayedAuditoryFeedback. DAF, delayed auditory feedback; dPreCG, dorsal precentral gyrus; IFG, inferior frontal gyrus; postCG, postcentral gyrus; SEM, standard error of the mean; SMG, supramarginal gyrus; STG, superior temporal gyrus; vPreCG, ventral precentral gyrus.

To control for articulation duration, we identified all no delay and 200-millisecond DAF trials of the same stimulus item that match in articulation duration (i.e., articulation duration difference is smaller than 10 milliseconds). We compared the neural responses for no delay and 200-millisecond DAF trials by performing a paired *t* test at each time point, and we corrected for multiple comparisons using FDR (q = 0.05) and marked the time intervals that show a significant difference ($p < 0.01$) for at least 200 consecutive milliseconds. We found that neural responses were enhanced in STG, SMG, dPreCG, and IFG for the 200-millisecond DAF condition even when articulation durations were nearly identical to the ones for the no delay condition. In vPreCG and PostCG, where responses are presumably motor in nature, there was no response enhancement for 200-millisecond DAF condition after controlling for articulation duration (**Fig 7A–7F, S6 Fig**).

To control for the amount of feedback delay, we split the 200-millisecond DAF trials into 4 groups based on articulation duration: 0 to 25 percentile, 25 to 50 percentile, 50 to 75 percentile, and 75 to 100 percentile. We compared the neural responses by performing 1-way ANOVA at each time point, corrected for multiple comparisons using and FDR test (q = 0.05) and marked the time intervals that show a significant difference ($p < 0.001$) for at least 200 consecutive milliseconds. We did not find a significant neural response amplitude enhancement for longer articulation durations in any of the regions (**Fig 8A–8F, S7–S12 Figs**). This result shows that while the neural response amplitude is enhanced as a function of delay

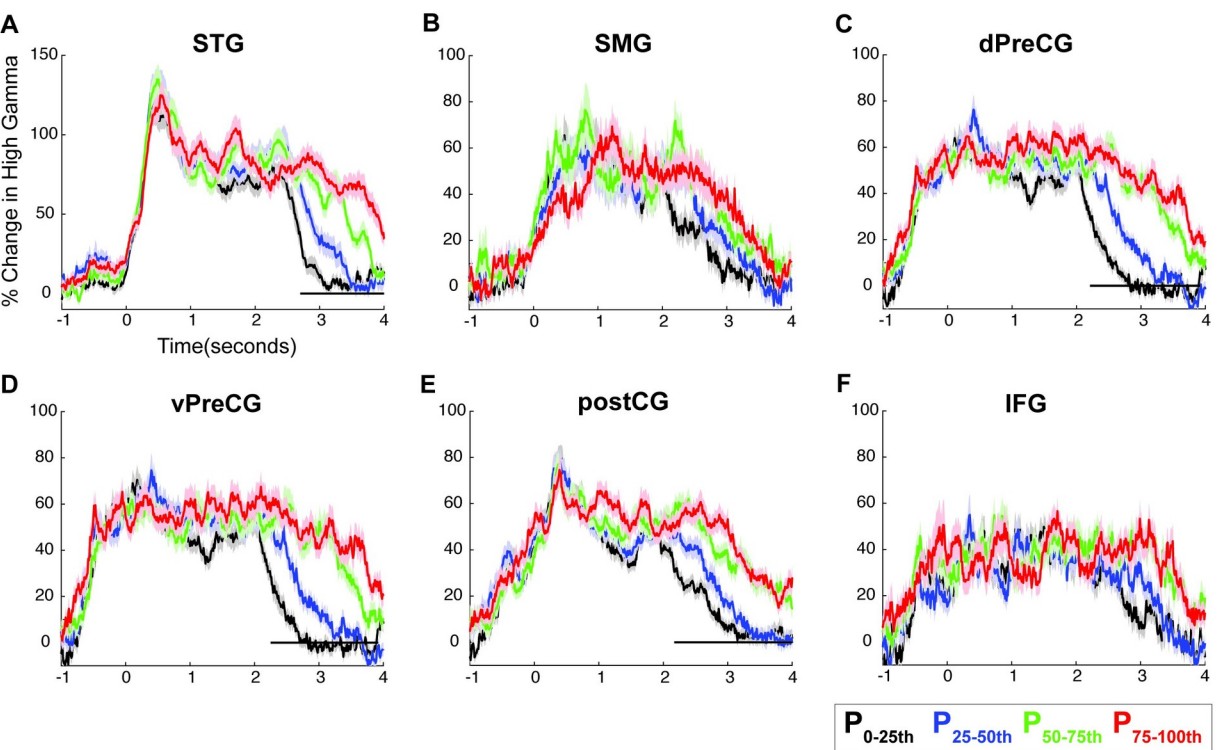

**Fig 8. Neural responses for DAF sentence reading task after controlling for delay condition. (A–F)** High gamma responses after 200-millisecond delay trials are split into 4 groups based on articulation duration. Activity locked to articulation of sentences is shown for 0 to 25th (black), 25th to 50th (blue), 50th to 75th (green), and 75th to 100th (red) percentiles of articulation durations in 6 different regions: STG, SMG, vPreCG, dPreCG, postCG and IFG. The underlying data can be found in https://github.com/flinkerlab/DelayedAuditoryFeedback. DAF, delayed auditory feedback; dPreCG, dorsal precentral gyrus; IFG, inferior frontal gyrus; postCG, postcentral gyrus; SEM, standard error of the mean; SMG, supramarginal gyrus; STG, superior temporal gyrus; vPreCG, ventral precentral gyrus.

condition, there is no such amplitude increase for longer articulations in the first several seconds during which all percentile conditions actually contain speech stimuli (i.e., the participant is still speaking). These complementary results eliminate the possibility that neural response enhancement is simply a motor phenomenon caused by longer articulation during DAF and provide strong evidence that it is due to auditory error processing in STG, SMG, and dPreCG.

## Discussion

Artificially slowing down speech when hearing one's own delayed voice provides a strong framework to investigate how auditory feedback influences the motor control of speech. Our study is one of the few electrophysiological investigations [35,36] and, to our knowledge, the only ECoG investigation of the underlying neural substrates. We compared the effects of DAF on producing short versus long speech segments by using word and sentence stimuli and showed that producing sentences during DAF had a stronger disruptive effect on speech. We used an unsupervised clustering algorithm (NMF) to determine auditory and motor regions involved in speech production and then identified 3 subregions of the speech network that are centrally engaged in the processing of auditory feedback: STG, SMG, and dPreCG. ECoG recordings provided us with the precise spatiotemporal evolution of feedback processing in these distinct regions. Neural responses were enhanced in amplitude and extended in duration for large delays reflecting the error signal caused by altered feedback and the subsequent longer

articulation. To dissociate the error signal from the effect of prolonged articulation, we followed different approaches in which we either use DTW algorithm to temporally align the neural signals with the participants' speech acoustics, or we control for articulation duration and the amount of feedback delay. We found that dPreCG showed response enhancement immediately after auditory cortex when speech fluency was strongly disrupted during production of sentences with DAF. These results highlighted dPreCG as a critical region for maintaining speech fluency when dynamic auditory feedback processing is required to produce longer utterances.

During speech production, the reafferent (perceived) auditory feedback is not immediately useful due to noise and delays in neural processing (axonal transmission, synaptic processes, etc.). According to predictive models of speech motor control, the brain must therefore rely on an internal estimate of the auditory feedback and use reafferent feedback only to correct this internal estimate. When an utterance is produced, an efference copy of the motor command is used to make a prediction of the current articulatory state (e.g., state of the vocal tract) and the subsequent sensory outcome. As long as there is no mismatch between the predicted and the reafferent feedback, the brain can rely on its internal estimate. However, when there is a mismatch, the brain generates an error signal to correct its internal estimate and the necessary motor commands to produce the intended utterance. In our paradigm, an artificially introduced delay generates a continuous mismatch between the predicted and the reafferent feedback. Specifically, the reafferent feedback matches the previous articulatory state rather than the current articulatory state. In this case, the issuing of new motor commands must be delayed, which would explain the slowing down of speech production. We found that producing single words with DAF elicited a slight slowing down effect and increased neural responses only in auditory but not in motor regions. However, when participants produced sentences with DAF, this longer and more complex stimulus elicited a much prominent slowing down effect and increased neural responses in auditory as well as motor regions.

Neural responses to DAF were enhanced in STG, SMG, and dPreCG, which are anatomically connected by one of the major language pathways known as the superior longitudinal fasciculus [37,38]. These regions are typically modeled as the main components of the dorsal stream for speech that is responsible for sensorimotor integration and auditory feedback processing [39]. In support of these theoretical models, clinical reports demonstrated that posterior STG and SMG damage are implicated in conduction aphasia [40], and patients with conduction aphasia are less affected by DAF [41], indicating the involvement of this regions in feedback processing.

Our analysis of the time course of responses revealed that response enhancement to DAF started in STG and closely followed by dPreCG providing further evidence for a functional correspondence between the 2 regions. dPreCG is a complex functional region implicated in auditory, motor, and visual speech processing [42–47]. It is known to be activated not only during speaking but also during passive listening suggesting a role in mapping acoustic speech features onto the corresponding articulatory movements. In an additional task with a subset of our participants who were presented with auditory speech stimuli, we also found activation in dPreCG verifying these previous findings (**S2 Fig, S2** Text). A possible control, which we were unable to complete in this study, would be to replay a recording of the participants' own delayed voice back to them [36] to completely eliminate the possibility of a purely sensory explanation for the observed response enhancement in dPreCG. But it is noteworthy that in addition to responding to auditory speech stimuli, dPreCG electrodes also respond strongly prior to speech during motor preparation. This prearticulatory activity taken together with the increased activity during delayed feedback suggests auditory error processing rather than a purely sensory explanation. We predict that dPreCG may be the hub for generating the

internal estimate of the speech output by processing both the efference copy and auditory error signals during speech production.

Interestingly, response enhancement in dPreCG during auditory feedback processing has never been reported previously. Studies that altered auditory feedback using pitch perturbation demonstrated enhanced responses in ventral parts of the precentral gyrus, which correlated with the compensatory vocal adjustments [12–14]. In our experiments, vPreCG did not show any response enhancement neither for word nor for sentence production with DAF. It could be the case that vocal control that requires pitch adjustments activate vPreCG, while vocal control that requires long-term maintenance of other prosodic features, such as tempo, rhythm, and pause, activate dPreCG. While our results ruled out the possibility that response enhancement in dPreCG during sentence reading is due to longer articulation duration, we did not have the necessary control tasks to test whether stimulus length or complexity drives this modulation. However, dPreCG is well known to be involved in movement planning and execution [48,49]. Previous studies reported increased activation in this region when participants produced complex syllable sequences, suggesting a role in the planning and production of long speech utterances with appropriate syllable timing [50]. Considering that dPreCG activation occurred only when participants produced sentences with DAF, we predict that it plays a critical role in maintaining prosody and speech fluency during the articulation of long utterances.

Our results showed that the maximal disruption of speech occurred at 200-millisecond feedback delay for both word reading and sentence reading tasks. Speech paradigms in previous DAF studies used various amounts of delays ranging from 25 to 800 milliseconds and consistently reported that the strongest disruption of speech occurred at 200-millisecond delay [18–20,51,52]. This time interval is thought to be critical for sensorimotor integration during speech production because it is of about the same order of average syllable duration. Given that the temporal distance between 2 consecutive stressed syllables is roughly 200 milliseconds, it has been suggested that delaying auditory feedback by this amount of time causes a rhythmical interference that results in the maximal disruption of speech fluency [53].

The dynamics of the cortical speech network can also provide further explanation for the maximal disruption of speech at 200-millisecond feedback delay. Previous ECoG studies showed that IFG is activated before articulation onset and remained silent during articulation, while motor cortex is activated both before and during articulation. These studies suggested that IFG produces an articulatory code that is subsequently implemented by the motor cortex and reported an approximately 200-millisecond temporal lag between IFG and motor cortex activation [54,55]. A feedback delay in the same order of this temporal lag likely interrupts propagation of the articulatory code from IFG to motor cortex, thereby disrupting speech. Unlike prior reports, we found sustained IFG activity throughout speech production; however, this was during DAF where sustained IFG recruitment may be necessary to support compensatory speech correction. The onset of IFG activation was seen in conjunction with dPreCG; however, sensitivity to DAF was seen at 2 distinct time periods with an early recruitment of dPreCG and much later involvement of IFG.

Behavioral paradigms that manipulate auditory feedback have been widely used for decades to understand speech motor control; however, the cortical dynamics underlying this process remained largely unknown. We elucidate the magnitude, timing, and spatial distribution of the neural responses that encode the mismatch between produced speech and its perceived feedback. Our results highlighted STG, SMG, and dPreCG as critical players in detection and correction of vocalization errors. Specifically, we find that dPreCG is a selective region that is recruited immediately when auditory feedback becomes unreliable and production more effortful, implicating it in auditory-motor mapping that underlies vocal monitoring of human speech.

## Supporting information

**S1 Fig. Neural responses before and after excluding trials with speech errors. (A–F)** High gamma responses for the DAF sentence reading in 6 different regions are shown. Responses before and after the exclusion of speech errors are shown in the left and right panels, respectively. Colors represent the various DAF conditions and shaded regions indicate SEM over trials. Black horizontal lines at the bottom of the graph indicate the time intervals when the neural responses diverged significantly across conditions. The underlying data can be found in https://github.com/flinkerlab/DelayedAuditoryFeedback. DAF, delayed auditory feedback; SEM, standard error of the mean.
(TIF)

**S2 Fig. Neural responses to a passive listening task in dPreCG. (A)** Seventeen dPreCG electrodes in 5 participants are shown on a template brain. Electrodes that show a significant response during passive listening of words are shown in red. **(B)** Average high gamma responses across the dPreCG electrodes is shown. Shaded regions indicate SEM over trials. The underlying data can be found in https://github.com/flinkerlab/DelayedAuditoryFeedback. dPreCG, dorsal precentral gyrus; SEM, standard error of the mean.
(TIF)

**S3 Fig. Sensitivity to DAF measured as the correlation between neural response and auditory error. (A)** Correlation between high gamma response and auditory error for each electrode is shown on the template brain. **(B)** Sensitivity to DAF measured as the correlation between high gamma response and delay versus the correlation between high gamma response and auditory error. The underlying data can be found in https://github.com/flinkerlab/DelayedAuditoryFeedback. DAF, delayed auditory feedback.
(TIF)

**S4 Fig. Sensitivity to DAF in each electrode calculated by linear regression.** A linear model was fit to describe the relationship between neural response and delay condition. For each electrode, slope of the fitted line was used as measure of sensitivity to DAF and shown on a template brain for **(A)** word reading and **(B)** sentence reading tasks. The underlying data can be found in https://github.com/flinkerlab/DelayedAuditoryFeedback. DAF, delayed auditory feedback.
(TIF)

**S5 Fig. Pairwise comparisons of neural responses to different delay conditions. (A–F)** Black horizontal lines indicate the time intervals when the neural response to the 2 delay conditions diverged significantly in 6 different regions of interest for at least 200 consecutive milliseconds (1-way ANOVA p<0.01 with FDR correction at q = 0.05). Divergence onset times are indicated with red text. The underlying data can be found in https://github.com/flinkerlab/DelayedAuditoryFeedback. FDR, false discovery rate.
(TIF)

**S6 Fig. Comparisons of neural responses after controlling for articulation duration.** Scatter plots show the averaged high gamma responses between 0 and 2 seconds for "no delay" versus "200-millisecond delay" conditions for in each electrode (black circles) in different regions. Only STG, SMG, and dPreCG showed significantly larger neural responses for 200-millisecond delay condition (paired $t$ test; STG: t = 5.6, $p = 2 \times 10^{-6}$, SMG: t = 2.2, $p = 0.04$, dPreCG: t = 2.43, $p = 0.02$, vPreCG: t = 0.39, $p = 0.7$, postCG: t = 0.86, $p = 0.4$, IFG: t = 2.03, $p = 0.06$). The underlying data can be found in https://github.com/flinkerlab/DelayedAuditoryFeedback.

dPreCG, dorsal precentral gyrus; SMG, supramarginal gyrus; STG, superior temporal gyrus.
(TIF)

**S7 Fig. Neural responses in STG after controlling for delay condition.** High gamma responses after 200-millisecond delay trials are split into 4 groups based on articulation duration are shown for each single electrode in STG. The underlying data can be found in https://github.com/flinkerlab/DelayedAuditoryFeedback. STG, superior temporal gyrus.
(TIF)

**S8 Fig. Neural responses in SMG after controlling for delay condition.** High gamma responses after 200-millisecond delay trials are split into 4 groups based on articulation duration are shown for each single electrode in SMG. The underlying data can be found in https://github.com/flinkerlab/DelayedAuditoryFeedback. SMG, supramarginal gyrus.
(TIF)

**S9 Fig. Neural responses in dPreCG after controlling for delay condition.** High gamma responses after 200-millisecond delay trials are split into 4 groups based on articulation duration are shown for each single electrode in dPreCG. The underlying data can be found in https://github.com/flinkerlab/DelayedAuditoryFeedback. dPreCG, dorsal precentral gyrus.
(TIF)

**S10 Fig. Neural responses in vPreCG after controlling for delay condition.** High gamma responses after 200-millisecond delay trials are split into 4 groups based on articulation duration are shown for each single electrode in vPreCG. The underlying data can be found in https://github.com/flinkerlab/DelayedAuditoryFeedback. vPreCG, ventral precentral gyrus.
(TIF)

**S11 Fig. Neural responses in postCG after controlling for delay condition.** High gamma responses after 200-millisecond delay trials are split into 4 groups based on articulation duration are shown for each single electrode in postCG. The underlying data can be found in https://github.com/flinkerlab/DelayedAuditoryFeedback. postCG, postcentral gyrus.
(TIF)

**S12 Fig. Neural responses in IFG after controlling for delay condition.** High gamma responses after 200-millisecond delay trials are split into 4 groups based on articulation duration are shown for each single electrode in IFG. The underlying data can be found in https://github.com/flinkerlab/DelayedAuditoryFeedback. IFG, inferior frontal gyrus.
(TIF)

**S1 Text. Speech error analysis.**
(DOCX)

**S2 Text. Passive listening task.**
(DOCX)

**S3 Text. Auditory error calculation.**
(DOCX)

**S4 Text. Linear regression analysis to calculate sensitivity to DAF.** DAF, delayed auditory feedback.
(DOCX)

**S5 Text. Pairwise comparison of neural responses.**
(DOCX)

## Acknowledgments

We thank Zhuoran Huang and Qingyang Zhu for their assistance in analyzing voice recordings of the participants.

## Author Contributions

**Conceptualization:** Muge Ozker, Adeen Flinker.

**Data curation:** Muge Ozker, Adeen Flinker.

**Formal analysis:** Muge Ozker, Adeen Flinker.

**Funding acquisition:** Muge Ozker, Adeen Flinker.

**Investigation:** Muge Ozker, Adeen Flinker.

**Methodology:** Muge Ozker, Werner Doyle, Orrin Devinsky, Adeen Flinker.

**Project administration:** Muge Ozker, Adeen Flinker.

**Resources:** Muge Ozker, Werner Doyle, Orrin Devinsky, Adeen Flinker.

**Software:** Muge Ozker, Adeen Flinker.

**Supervision:** Muge Ozker, Adeen Flinker.

**Validation:** Muge Ozker, Adeen Flinker.

**Visualization:** Muge Ozker, Adeen Flinker.

**Writing – original draft:** Muge Ozker, Adeen Flinker.

**Writing – review & editing:** Muge Ozker, Adeen Flinker.

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
