## [Editor Report · Decision Letter 0]

24 Aug 2021

Dear Dr Ozker, 

Thank you for submitting your revised manuscript entitled "Cortical network underlying speech production during delayed auditory feedback" for consideration as a Research Article by PLOS Biology.

Your new version and your response to reviewers have now been evaluated by the PLOS Biology editorial staff, and I am writing to let you know that we would like to send your submission out for external peer review.

Please re-submit your manuscript within two working days, i.e. by Aug 26 2021 11:59PM.

Kind regards,

Gabriel Gasque

Senior Editor

PLOS Biology

ggasque@plos.org

---

## [Decision Letter · Decision Letter 1]

1 Oct 2021

Dear Dr Ozker,

Thank you for submitting your revised Research Article entitled "Cortical network underlying speech production during delayed auditory feedback" for publication in PLOS Biology. I have now obtained advice from the original reviewers and have discussed their comments with the Academic Editor. 

Based on the reviews, we will probably accept this manuscript for publication, provided you satisfactorily address the remaining points raised by the reviewers. Please also make sure to address the following data and other policy-related requests:

1) Title

We would like to suggest a title that would be more appealing to a broader audience. We recommend: 

1.a) A cortical network processes auditory error signals during human speech production to maintain fluency.

Or if you wish to highlight the role of the dorsal precentral gyrus, the title could be:

1.b) A cortical network including the dorsal precentral gyrus processes auditory error signals during human speech production to maintain fluency.

Or 

1.c) A cortical network underlies human speech production during delayed auditory feedback.

We favor option 1.a because of its accessibility to non-specialists. However, we would be happy to work with you on an alternative if our recommendation is not accurate or misrepresents your findings.

2) Blurb

Please provide a blurb which (if accepted) will be included in our weekly and monthly Electronic Table of Contents, sent out to readers of PLOS Biology, and may be used to promote your article in social media. The blurb should be about 30-40 words long and is subject to editorial changes. It should, without exaggeration, entice people to read your manuscript. It should not be redundant with the title and should not contain acronyms or abbreviations. For examples, view our author guidelines: https://journals.plos.org/plosbiology/s/revising-your-manuscript.

3) Ethics:

3.a) Please include information about the form of consent (written/oral) given for research involving human participants. If oral, please explain why.

3.b) Please indicate if your experiments were conducted according to the principles expressed in the Declaration of Helsinki or any other specific national or international ethical guidelines.

4) Data:

4.a) You may be aware of the PLOS Data Policy, which requires that all data be made available without restriction: http://journals.plos.org/plosbiology/s/data-availability. For more information, please also see this editorial: http://dx.doi.org/10.1371/journal.pbio.1001797

Note that we do not require all raw data. Rather, we ask for all individual quantitative observations that underlie the data summarized in the figures and results of your paper. For an example see here: http://www.plosbiology.org/article/info%3Adoi%2F10.1371%2Fjournal.pbio.1001908#s5

These data can be made available in one of the following forms:

Regardless of the method selected, please ensure that you provide the individual numerical values that underlie the summary data displayed in the following figure panels: Figures 1BDE, 2ACD, 3BCDG, 4A-H, 5A-F, 6A-F, 7A-F, S1A-F, S2B, S3B, S5A-F, and S6.

4.b) Please also ensure that each figure legend in your manuscript includes information on where the underlying data can be found and that your supplemental data file/s has/have a legend.

4.c) Please ensure that your Data Statement in the submission system accurately describes where your data can be found.

We expect to receive your revised manuscript within two weeks. 

*Published Peer Review History*

*Early Version*

Sincerely,

Gabriel Gasque, Ph.D.,

Senior Editor,

ggasque@plos.org,

PLOS Biology

Reviewer remarks:

Reviewer #1, Edward Chang: This manuscript has been significantly improved since the first submission. The responses from the authors are very helpful and have addressed most of my concerns. The newly added Figures. 6 & 7 provided important new evidence to support the sensitivity on auditory error in the dorsal precentral gyrus. The additional analysis included in the supporting information is also very helpful for me to understand the experimental details and controls. I appreciate the efforts the authors made to improve this work. I think the results presented in the current version are convincing and the work adds important information to the research of feedback processing in speech production. I have some remaining questions, but I believe they are all addressable.

(1) Fig. 3, is it possible to provide an example of a single electrode's activity, similar to Fig. 1D, since this is the first time to show the results for sentence reading? This also helps to illustrate the modulation of delay on the single electrode level, e.g. whether each delay change caused a similar amount of change in activity; how the activity evolves over time, etc.

(2) Page 14, last sentence: the statement regarding "articulating longer and more complex speech stimuli" is not as grounded as it sounds (similar statements appear in multiple places). While I understand the results from sentence reading and its contrast to word reading, it is unclear whether it is the length or the complexity that drives the modulation in dPreCG. One possible control is to ask the subject to repeat the same word (or syllables) multiple times so that it matches the length of a sentence, or to use words with complex structure while keeping the number of syllables the same. Ideally one could test these in experiments, but I understand the difficulty with intracranial recordings. So if these data are not available, some discussions about the limitations need to be added. 

(3) Fig. 6 & 7, while these are important new results, it is hard to know how much variation there is across individual electrodes. Is it possible to show a scatter plot (or other reasonable formats) to summarize the comparison for all the electrodes? Presumably, a window with the largest effect can be selected to quantify the difference across conditions. One potential concern for Fig. 7 is that the activity of some electrodes is positively correlated with articulation duration, and the activity of other electrodes is negatively correlated. When averaged together, it appears that there is no modulation. 

(4) In the abstract, "response enhancement in dPreCG occurred only when subjects profoundly slowed down their speech" seems misleading. This implies that dPreCG activity is modulated by articulation (speed/duration), which is the opposite of what the main conclusion is. In fact, the authors have shown that given the same delay, articulation duration does not change the dPreCG activity level. Please consider changing the expression.

Reviewer #2: I think overall, the authors have done a good job addressing most of my comments.

One of my concerns still remains about auditory controls. While the authors did show some speech playback data as a supplement for their revised manuscript, there are still some problems since it was 1) not the subjects own voice, and 2) did not test the effects/distortions from feedback delays. I think its probably fine, but may need some more caveats in the discussion.

Reviewer #3: Thank you for revising your paper to respond to my comments. I think this paper is ready to be published in Plos Biology.

---

## [Editor Report · Decision Letter 2]

8 Nov 2021

Dear Muge,

Thank you for submitting your revised Research Article entitled "A cortical network processes auditory error signals during human speech production to maintain fluency" for publication in PLOS Biology. I have discussed your response to reviewers and your revision with the Academic Editor. We think we are almost there, but would like you to revise one more time to address the following points:

1) All your new analyses included in the response to reviewers should be incorporated into the manuscript, either in the main text or, at least, as supporting information (but cited within the main text), in the following way:

1.a) Together with the Academic Editor, we think that the first analysis requested by reviewer 1 is very nice and adds value and should definitely be included either within Figure 6 or the supplemental. materials. 

1.b) We think that you probably misunderstood the second point (regarding Figure 7) as we think the reviewer's concern is about channels that come from areas that *don't* show a difference; the electrodes that are shown are from dPreCC - the electrodes for vPreCG and PostCG are those that should be shown. Therefore, would suggest that you should either plot these areas (in which case you may as well show all electrodes from all areas and put it all in the supplemental material - this would be our preferred option), or do some sort of scatter plot at a fixed (maximal) time point as you did for Figure 6. The worst case scenario is that you need to rephrase to say that in the motor areas responses were more heterogenous such that on average there was no difference (if reviewer 1's point is correct and some show positive and others negative effects). If there is something we misunderstood as to why you can't do this, then feel free to contact me via email to clarify, and we can discuss further.

2) Please change the title in the manuscript itself to the one we have already agreed: "A cortical network processes auditory error signals during human speech production to maintain fluency."

3) I have some confusion regarding the figure numbers in this version. Old figure 6 seems to be now new Figure 8 and not 7, as you have stated in the figure legends. Could you please correct and/or clarify?

4) Could you please update your GitHub repository to include data for Figures 1E and 3E (or clarify why these data are not needed/provided)?

We expect to receive your revised manuscript within two weeks. 

*Published Peer Review History*

*Early Version*

Sincerely,

Gabriel Gasque, Ph.D.,

Senior Editor,

ggasque@plos.org,

PLOS Biology

---

## [Editor Report · Decision Letter 3]

24 Nov 2021

Dear Muge,

On behalf of my colleagues and the Academic Editor, Jennifer Bizley, I am pleased to say that we can in principle accept your Research Article "A cortical network processes auditory error signals during human speech production to maintain fluency" for publication in PLOS Biology, provided you address any remaining formatting and reporting issues. These will be detailed in an email that will follow this letter and that you will usually receive within 2-3 business days, during which time no action is required from you. Please note that we will not be able to formally accept your manuscript and schedule it for publication until you have any requested changes.

**IMPORTANT: As you address these issue, please also make sure to include in the Supporting Figure Legends where the underlying data can be found: "The underlying data can be found in https://github.com/flinkerlab/DelayedAuditoryFeedback" 

PRESS

Sincerely, 

Gabriel Gasque, Ph.D. 

Senior Editor 

PLOS Biology

ggasque@plos.org